# Salmonidae Genome: Features, Evolutionary and Phylogenetic Characteristics

**DOI:** 10.3390/genes13122221

**Published:** 2022-11-27

**Authors:** Artem P. Dysin, Yuri S. Shcherbakov, Olga A. Nikolaeva, Valerii P. Terletskii, Valentina I. Tyshchenko, Natalia V. Dementieva

**Affiliations:** 1Russian Research Institute of Farm Animal Genetics and Breeding-Branch of the L.K. Ernst Federal Research Center for Animal Husbandry, Pushkin, 196601 St. Petersburg, Russia; 2All-Russian Research Veterinary Institute of Poultry Science-Branch of the Federal Scientific Center, All-Russian Research and Technological Poultry Institute (ARRVIPS), Lomonosov, 198412 St. Petersburg, Russia

**Keywords:** family Salmonidae, rediploidization, whole genome duplication, phylogeny

## Abstract

The salmon family is one of the most iconic and economically important fish families, primarily possessing meat of excellent taste as well as irreplaceable nutritional and biological value. One of the most common and, therefore, highly significant members of this family, the Atlantic salmon (*Salmo salar* L.), was not without reason one of the first fish species for which a high-quality reference genome assembly was produced and published. Genomic advancements are becoming increasingly essential in both the genetic enhancement of farmed salmon and the conservation of wild salmon stocks. The salmon genome has also played a significant role in influencing our comprehension of the evolutionary and functional ramifications of the ancestral whole-genome duplication event shared by all *Salmonidae* species. Here we provide an overview of the current state of research on the genomics and phylogeny of the various most studied subfamilies, genera, and individual salmonid species, focusing on those studies that aim to advance our understanding of salmonid ecology, physiology, and evolution, particularly for the purpose of improving aquaculture production. This review should make potential researchers pay attention to the current state of research on the salmonid genome, which should potentially attract interest in this important problem, and hence the application of new technologies (such as genome editing) in uncovering the genetic and evolutionary features of salmoniforms that underlie functional variation in traits of commercial and scientific importance.

## 1. Introduction

Due to their great diversity, morphology, ecology, genetics, and genomics, as well as their higher economic value, biology, and culture, fish are a very interesting group of animals. In this regard, more and more intensive efforts are now being devoted to the development of aquaculture. Efficient fish farming makes it possible to produce large volumes of high-quality products, which contribute to the conservation of natural populations of valuable fish species [1,2]. One of these interesting species of fish is the *Salmonidae* family, consisting of salmon, trout, char, grayling, whitefish, and their relatives, which has a scientific, social, and economic value that is unique among fish [3]. They are mainly known for performing some specific key functions in ecology, such as bringing marine nutrients to freshwater ecosystems [4].

Combined with their tetraploid origin, life history diversity, and rate of diversification, they have generated significant interest from the research community. Equally important, salmon is considered an excellent source of nutrients such as amino acids, lipids, vitamins, and minerals [5,6,7,8,9,10,11,12,13]. The problem remains that many salmon populations are declining, and great efforts are directed toward their conservation, especially in relation to anthropogenic factors [14]. Salmonids include 70 to 200 species with a wide variety of adaptations and life cycle strategies [15]. Salmon aquaculture and fisheries (mainly Atlantic salmon *Salmo salar* L. and *Oncorhynchus* spp.) play an important role in the economic and/or food security of several countries, accounting for 7.2/16.6% of all fish traded in terms of weight ratio/price [16].

Fortunately, genomics has given us new tools for addressing fundamental fishery management concerns, including stock recognition, population structure, and adaptive responses to ecological change [17,18,19]. The application of comparative genomics, molecular cell biology, and in situ hybridization of sequences and chromosomes using model and non-model fish species considerably helps our understanding of gene function, evolution, speciation, selection and adaptation, and species diversity [17]. The identifying of SNP markers by NGS has expanded the capacity to trace fishing resources or products back to the original site, allowing regulations for some industrially useful fish species to be enforced [20]. The Japanese pufferfish *Fugu rubripes* was the first to have its whole genome sequenced [21]. Since the first main commercial species, the Atlantic cod, was sequenced in 2011, aquaculture and fisheries genomics have made substantial progress [22]. Over 200 fish genomes have been sequenced and made accessible in public repositories since the invention and enhancement of massively parallel sequencing technology began in about 2005 [23]. Population genomic analysis using RAD sequencing or genome resequencing has been implemented for a few commercially important species, such as Asian sea bass [24], European sea bass [25], and Atlantic cod [22,26], revealing the genetic basis of fisheries-driven evolution and the possible repercussions of changes in the environment. High-quality genomes for numerous fish species have recently been released, providing insights into the molecular basis of evolution and biology [27,28,29,30]. A large amount of knowledge has been gained by sequencing the teleost fish genome. For example, the spotted gar (*Lepisosteus oculatus*) genome has shown how immunity evolution, mineralization, and development can be mediated by Hox, ParaHox, and microRNA genes [24]. The genetic foundation of polyploidy and parallelism between basal and fish-derived lineages to acquire duplications has been shown by genome sequencing of ancient fish lineages such as the starfish (*Acipenser ruthenus*) [31]. The Genome 10 k project, for instance, was established in 2009 by a partnership of biologists and genomic researchers to sequence the full genomes of 10,000 vertebrate species, including around 4000 fish species, in order to comprehend vertebrates evolution and conserve many endangered species [32,33,34].

Returning to the specific object of our discussion, the fish of the salmon family, we flesh out this discussion regarding them. The NCBI database currently provides data on 41 genomic assemblages of salmoniforms, which may indicate a dynamic trend in their study [35]. The high phenotypic diversity among salmonids has provided an excellent research framework for understanding adaptive divergence and ecological speciation [36,37] and potentially contributed to the WGD in their common ancestor, an event known as Ss4R [38], about 95 Ma ago [39,40]. Thus, one of the most remarkable features of the evolutionary history of salmonids is their autopolyploid origin [41,42]. This makes salmon an ideal organism for studying the consequences of the genome and gene duplications [43], which are thought to have played a key role in creating the gene diversity and functional specialization found in modern vertebrates [44,45]. In this regard, the genome of the Atlantic salmon (*Salmo salar*) was chosen as the reference sequence for all salmonids due to its importance in the aquaculture industry and because so much research on this species had previously been undertaken at the genomic level [46]. Significant progress in understanding salmon biology has been made by sequencing two salmon genomes, as well as the genome of the northern pike *Esox lucius*, a sister line that has not undergone salmon-specific whole genome duplication [47]. By taking into account these discoveries, it can be concluded that genome reconstruction and comparative genomic analysis of various fish species have become a successful approach to understanding their evolution. In this regard, in our work, we present an updated and comprehensive review of recent achievements in the genomics of salmonids and highlight the key results of various comparative genomic approaches for their study.

## 2. Characterization of the Salmon Genome

### 2.1. WGD

As noted above, one of the most notable features of the evolutionary history of salmonids is their autopoliploid origin [48]. It has been suggested that whole genome duplication (WGD), which characterizes all inherited salmonids [48], occurred, according to various estimates, from 58–63 to 88–103 million years ago [40], resulting in a single tetraploid ancestor in salmonids, whose autotetraploidization has been termed an ‘autopolyploidization’ (Ss4R) event [38,49]. Rediploidization occurs after all WGD occurrences, resulting in additional genomic diversity [50].

After WGD in the same species (autopolyploidization), rediploidization entails a change during meiosis from polyvalent (tetraploid inheritance) to bivalent chromosomal mating (diploid inheritance) [38,51]. As a result, recombination between the four alleles stops, causing sequence divergence between duplicated genes on distinct chromosomes (ohnologues) [50,51]. This, in turn, creates new pathways of functional evolution, unlike before WGD [50,52,53]. However, genetic studies of organisms with relatively recent WGD can be hampered by the inability to distinguish alleles and sequences from the same chromosome from alleles and sequences from a duplicated chromosome [42]. Fortunately, approaches using gamete manipulation, whole-genome sequencing, and long-read sequencing have improved the ability of researchers to characterize duplicated regions. Linkage mapping to haploid and double haploid has facilitated the analysis of duplicated regions in salmonids [54,55]. Furthermore, long-read sequencing methods have enabled the assembly of large genomes with complicated duplication histories [56]. These technical developments have improved our understanding of the genomic architecture of species that have dynamically extended to multiple species after ancestral WGD in lineages such as salmonids [38,57,58].

Mutations that promote preferred bivalent mating during meiosis, such as structural rearrangements and mobile element insertions, are required for rediploidization [38,59]. The same rediploidization process does not occur in allopolyploids, where WGD emerges after hybridization between different species, proving immediate preferential bivalent mating of subgenomes originating from each parental species [60,61], but it can possibly occur whenever sequence similarity is sufficient to generate polyvalent pairs, as in segmental allopolyploids [62]. It is widely believed that WGD promotes evolutionary diversification through mechanisms that are still not fully understood [63]. Some authors have suggested that WGD, which is salmonid-specific, is followed by radiation exposure [64,65]. Autotetraploidization involves the spontaneous doubling of all chromosomes distinct from the other major WGD classes, allotetraploidization, which involves hybridization between different species.

It has been observed that interspecific hybridization in salmonids is a frequent phenomenon. Some publications describe hybrids between Arctic char *S. alpinus* and *S. malma* [66], Dolly Varden and brook trout *S. confluentus* [67], *S. confluentus* and *S. fontinalis* [68], Arctic char and river trout [69], mottled char *S. leucomaenis* and *S. levanidovi* [70], mallow and white-spotted char [71]. As a rule, hybrids might be identified accordingly to their external morphology [72,73,74]. Some researchers consider interspecific hybridization in loaches as a consequence of the formation and existence of so-called contact zones between species during post-glacial colonization [69]. Other authors attribute cases of interspecific hybridization to the anthropogenic transformation of fish habitats or the invasion of alien species [75,76,77,78,79]. Here, there is no reason to say that the system is practically in a pristine state [80]. More recently, Gruzdeva et al. (2018) used genetic marker analysis to find hybrids formed by female white-spotted char and male northern Dolly Varden [81].

Although the impact of WGDs on long-term processes such as organism diversification is unknown [82,83], findings demonstrate that WGDs cause numerous problems during mitosis and meiosis [84]. These concerns are reflected in the fact that most neopolyploid species revert to diploidy rapidly following duplication [85]. Campbell et al. (2019) reported that gene conservation is a prominent result of WGD in salmonids. Although approximately 10% of the rainbow trout genome is inherited tetrasomally, ~28% of conserved ohnologue pairs are found in tetrasomal regions, suggesting that tetrasomal inheritance inhibits neofunctionalization and molecular adaptation. Furthermore, both the function and expression of tetrasomal ohnologues are preserved [57].

Early research has revealed that the most common outcome of a duplicated gene is for it to become dysfunctional and lost [86]. Although most ohnologists’ ultimate fate is gene loss following WGD, some ohnologists remain [87,88]. Genes predisposed to haploinsufficiency may be more likely to be conserved as functional duplicates to counteract the potential negative consequences of mutations in a single copy, i.e., the dangerous duplicate hypothesis [89,90]. However, it has been suggested that the selection for duplicate conservation and prevention of haploinsufficiency is weak, limiting the potential of this explanation for duplicate conservation [91,92]. Nevertheless, duplicates involved in immune function, cell cycle regulation, transcriptional control, and cell signaling pathways are conserved during vertebrate evolution [93,94,95]. However, the enriched GO term analysis presented by Campbell et al. (2019) found limited evidence for genes associated with such functions in tetrasomal regions of the genome [57].

The underlying determinants of delayed rediploidization remain a mystery. Rediploidization may be severely inhibited in places with delayed rediploidization due to negative implications for specific genes or favorable benefits of keeping genes as tetraploid. Gene set enrichment and gene expression analysis, for example, may not resolve such possibilities but can give hints for comprehending what appears to be a complicated process. Consequently, selection for conservation and loss of duplicated genes is probably multifaceted when delayed rediploidization is involved [96]. For Ss4R, it is assumed that selection has maintained tetraploidy in SSR regions for tens of millions of years [97]. It’s also worth wondering why a second round of rediploidization was permitted. Because it is inextricably linked to the evolutionary origin of salmonid subfamilies, it is possible that the events leading to speciation coincided with a reduction in effective population size, reducing the efficiency of selection while also introducing the adverse effects of rediploidization [38,39,64]. Perhaps novel selected forces linked with early species diversification (e.g., related to the earliest formation of anadromy) affected selection during rediploidization and anthropogenic divergence in other ways owing to impacts on individual genes [98]. Long-read assemblages containing all salmonid species will provide a mechanistic understanding of the relationship between mobile element evolution, speciation, and lineage-specific rediploidization [99].

### 2.2. LORe and AORe

Previous research on salmonid fish has revealed that rediploidization happened at various stages of development for distinct parts of the genome following Ss4R, which dates back 88–103 million years [61]. After previous occurrences at the root of vertebrate [31] and bony fish evolution, Ss4R is the fourth WGD in salmonid evolutionary history [31,39,58,59,61]. Variable degrees of sequence divergence among large synthetic ohnologue blocks reflect the different times of rediploidization for duplicated regions left from Ss4R [38]. Although rediploidization occurred in the ancestors of all modern salmonids in large genomic regions, including several complete chromosomes, speciation occurred before rediploidization was completed in several large genomic segments [70]. Consequently, some duplicated chromosomal arms have very high sequence similarity (>95%) in all salmonid species [38,100,101,102] and experience anthropological divergence independently in three subfamilies of salmonids that diverged about 50 million years ago [70]. This process has been termed “lineage-specific ohnological resolution” (“LORe”) and is thought to be possible whenever an evolutionary transition from multivalent to bivalent mating [61,70].

Late rediploidization and LORe are crucial for phylogenetic reconstruction [70,100]. If global WGD expectations are not taken into account, LORe can easily be mistaken for lineage-specific (e.g., tandem) gene duplication during phylogenetic analyses [70]. In terms of the evolutionary significance of WGD events, it has been hypothesized that LORe promotes lineage-specific adaptation [70] and provides a viable explanation for the commonly observed temporal delays between a WGD event and subsequent species or phenotypic diversification regimes [103,104,105]. Many authors have noted the potential relevance of late rediploidization and LORe in divergent species after the discovery of these events in salmonids [40,64,104,105,106]. Despite the fact that salmonids are a well-established research system, many researchers’ understanding of rediploidization discoveries in this group of fishes remains fragmented as a result of whole genome sequence data and/or insufficient phylogenetic resolution in prior reconstructions.

Delayed rediploidization and LORe have yet to be unequivocally demonstrated beyond salmonids but have been proposed to track WGD events specific to bony fish (Ts3R) and WGD events at the base of vertebrates [70,71,106,107]. Several writers have shown that delayed and nested models of species diversity in the aftermath of WGD accord with LORe [64,104,105,107,108]. Given the large and growing number of high-quality genomic sequences in various eukaryotic lineages with a history of WGD, Gundappa et al. (2022) adapted their phylogenomic approach to rediploidization results for WGD events of comparable or earlier age for Ss4R, using synteny/collinearity to distinguish ohnologues from other duplicate genes. Because of the large evolutionary distances involved, genome alignment approaches would be inappropriate for studying earlier WGD events like Ts3R. The complex LORe patterns characteristic of large sections of the salmonid genome strongly violate the usual assumption for such methods that ohnologue divergence begins at a particular branch of the species tree [39].

To enhance the ability to reconstruct rediploidization dynamics during salmonid evolution, Gundappa et al. set out to reconstruct the rediploidization process after Ss4R and its results with significantly increased genomic and phylogenetic resolution compared with earlier studies, for which they created a high-quality genome sequence of Danube salmon (*Hucho hucho*) and developed a genome alignment approach to capture Ss4R-matched regions in several species. They sequenced the genome of a salmonid species with a highly interesting phylogenetic position and devised a whole-genome alignment method to catch ethnographic areas in genome assemblies previously released for many salmonid species. Using phylogenetic approaches, they were able to recreate whole-genome rediploidization dynamics, capturing two major waves of rediploidization as well as intricate histories of lineage-specific ohnologue divergence, the complexity of which rises with speciation history. The authors created multispecies alignments for a priori defined blocks of syntenic ohnologues left over from Ss4R [38] in two regions of the *Hucho hucho* genome where rediploidization was either ancestral to all salmonids (AORe regions with ancestral resolution [70]) or occurred after the separation of *Salmoninae* and *Thymallinae*. The alignments they obtained included species from *Salmoninae* and *Thymallinae* as well as a representative of *Esociformes*, a sister lineage of salmonids that diverged to Ss4R [38,40,70]. Their predecessor studies showed that about 25% of the *Salmoninae* genome underwent rediploidization after separation from *Thymallinae* [70,108].

Lien et al. (2016) showed that both tissue expression bias [38] and rediploidization time are important factors in ohnologue regulatory divergence in Ss4R ohnologues, with duplicates in SSR regions showing a higher correlation in tissue expression than AORe regions [70]. The results obtained by Gundappa et al. support these findings and show that expression level is an additional factor to be considered in the evolution of ohnologue regulation, thereby capturing differences in functional enrichment between ohnologues and singletons, which is consistent with past studies [109,110,111,112] but, in contrast to previous work [57,70], indicate general deviations in functional enrichment between Ss4R ohnologues with different rediploidization ages and suggest enrichment of a small number of unique functions in ohnologues from regions with different rediploidization ages [39].

In an earlier Ss4R study, all rainbow trout ohnologues classified as belonging to SSR regions were considered tetrasomic [113,114]. Gundappa et al. (2022) emphasize that their results require caution when confusing delayed rediploidization with no rediploidization [39]. As a result, tetrasomal heredity across many millions of years in rainbow trout areas designated as tetrasomal seems implausible [57]. Sequence similarity analysis using reference genomes by Blumstein et al. (2020) showed that all species possessed traits of residual tetrasomal heritability in seven homeologous pairs. By using linkage maps, the same seven homeologous pairs were found to be tetrasomal in *Oncorhynchus* [115,116], *Salvelinus* [117,118], *Salmo* [70], and probably *Coregonus* [119], which strongly suggests that tetravalent meiosis can and do form between these homeologs in all species examined to date. For several karyotypes, the evidence for residual tetrasomy differed between the linkage map and genome methods in Blumstein et al. (2020), with one categorized as tetrasomal in linkage mapping studies but not in genome analysis, and another categorized as tetrasomal in genome analysis but not in linkage maps [101]. This finding suggests that the frequency of diploidization in *Salvelinus* differs compared to other salmonids for this homeologous pair. A study of Arctic char found numerous duplicate markers in this karyotype [97]. The discovery that the karyotype is not tetrasomic, on the other hand, is most likely owing to differences in determining the level of residual tetraploidy between linkage mapping and genome assembly techniques. Linkage mapping in *Oncorhynchus* and *Salvelinus* consistently reveals the presence of tetrasomal inheritance [118,120], but genome analysis by Blumstein et al. (2020) and by Campbell et al. (2019) for rainbow trout found that this karyotype exhibits similarity to intermediate sequences consistent with disomic homeologs [59,101].

The last two genomes in the cell are usually different enough to split into two sets of bivalents during meiosis, which removes the incompatibility in mating between the hybridizing species before WGD [121]. Conversely, autotetraploidization results in four sets of chromosomes that initially diverge randomly during meiosis after WGD; bivalent mating must be restored before duplicate genes created by WGD can go beyond the allelic state [38,70,122]. In lake whitefish, aneuploidy has been documented across populations and historical circumstances [123,124]. Results from Blumstein et al. (2020) show that ambiguity in homologous relationships persists for at least five chromosome arms in all three coregonins [101]. This degree of ambiguity was much higher than that documented by Sutherland et al. (2017) in *Salmo*, *Oncorhynchus,* and *Salvelinus*, where there were only two ambiguities in these groups [117]. Whitefish appears to have many relatively small acrocentric chromosomes [125], some of which contain large numbers of duplicated loci, making it difficult to construct linkage maps compared with other salmonids [102,126].

Sutherland et al. (2017) investigated the history of fusions in various *Oncorhynchus* species and discovered that most species have numerous species-specific fusions (e.g., 17 species-specific fusions in pink salmon). Sutherland et al. (2017), on the other hand, researched only one species from the genera *Coregonus* and *Salvelinus* since that was all that was available at the time of publishing, and none from *Thymallus* [117]. The physiological consequence of proposed fusion histories is yet unknown, although it is a significant problem in distinguishing *Thymallus*, *Salvelinus*, and *Coregonus* spp. from many other salmonids. When high-quality genomes for a growing amount of salmon species become available, the accuracy with which it is possible to uncover changes in genome content and structure after whole genome duplication increases significantly. Additional data from genome sequencing studies, such as the European whitefish genome of De-Kayne et al. (2020), should allow for crucial additional research assessing genomic events and organization in species with distinct merging histories [127]. Using the northern pike’s chromosome numbering, these authors identified pairs of homeologous whitefish scaffolds and their associated ancestral chromosome. This should make future comparisons across the salmon family easier (as Blumstein et al. (2020) did) to examine the separate process of rediploidization in distinct salmon lineages. They also discovered a number of whitefish scaffolds with no known homeologs and demonstrated that some of these areas formed through the fusion of sequentially identical regions [128]. Diversity studies in salmoniforms are of essential scientific importance in order to comprehend the processes that produce and support such diversification, as well as to assist the conservation of this varied group. It will especially help to understand the genetic basis of their adaptability, including determining the degree of parallelism. In this regard, Table 1 reflects key data on the previously studied salmonid genomes (Table 1).

Many salmonid species have documented instances of residual tetrasomy and enhanced sequence similarity between homeologous chromosomes, implying that certain rediploidization might remain [38]. Although some species-specific differences in residual tetrasomal regions have been reported, tetraploidy appears to persist among salmonid species in the 7–8 homeologous pairs of chromosomes [117]. Although the evolution of the European grayling karyotype is otherwise comparable to that of most salmonids, such evolution is also observed in the European grayling genome assembly based on common linkage maps in the case of residual tetrasomal regions [129]. The evolutionary importance of persistent residual tetrasomy is unknown, but its existence in the European grayling ancestral genome demonstrates that tetrasomy is autonomous of the chromosomal fusions seen in other salmonids [38,125] rather than favors another cause. Therefore, we have elucidated the key characteristics of salmonid karyotypic evolution.

### 2.3. Mobile Elements in the Genome

Mobile elements could also play a key role in genome evolutionary processes [128]. Furthermore, these components may be significant in the processes of rediploidization, causing sequence divergence that would split homologs. Mechanistically, rediploidization involves the proliferation of mobile elements in the genome, which cause rearrangements that terminate polyvalent meiotic pairs, limiting anthropologic divergence [130]. Because bursts of mobile element activity are known throughout salmonid evolution [38], lineage-specific mobile element proliferation may be causally related to delayed lineage-specific rediploidization.

Unfortunately, the current generation of salmonid genomes does not allow such ideas to be tested because of their poor representation of mobile elements and genomic regions showing very recent rediploidization. In particular, it may be of interest to compare classes of mobile elements between organisms with very different genomic rearrangements, for example, between European grayling and Atlantic salmon. For example, Sävilammi et al. (2019) found that retrotransposons (class I mobile elements) are more abundant compared to DNA transposons (class II mobile elements) with a higher abundance of 1.7 and 1.3 times in the genomes of the European grayling and Atlantic salmon, respectively [129]. Differential accumulation of mobile elements between lineages may play a significant role in genome evolutionary processes, but the complexity of the underlying reasons for such differences must be explored to determine it. Elements specific to Atlantic salmon include DNA transposons, which together encompass 3.57% of the Atlantic salmon genome but are completely absent in European grayling. These mobile elements are one of the most prevalent types of mobile elements in salmonids, and they are thought to play a significant role in Atlantic salmon rediploidization [38]. The Copia-12 retrotransposon, which belongs to the Copia retrotransposon superfamily and has recently been proposed to have a role in chromosomal diversity and speciation in other bony fish, is one example of an Atlantic salmon-specific element [128].

*Tc1-Mariner* transposons, for example, *DNA hAT* transposons, are cut-and-insert elements with transposition processes that may actively drive genomic alternations as well as indirect pathways to make homologous recombination copies of elements [131]. The concentration of a specific mobile element in one of the two species could be interpreted as evidence of lineage-specific mobile element functioning. Understanding of mobile elements in the genome can give unique information to inspire future studies into the molecular mechanisms of these various genome evolutionary changes. More sensitive techniques, including such recurrent sequence grouping, may be employed to detect them since they may allow for a more accurate assessment of repeating element content [132]. Moreover, they may allow for a better understanding of repeat community structure and identification of key elements using network approaches [133], allowing for a more detailed investigation of repeat distribution dynamics among salmonids. It has been suggested that periods of impaired-purifying selection occurring in bottlenose populations is necessary to perpetuate the deleterious effects of chromosomal rearrangements [134]. Although chromosomal evolution may be initially chaotic, the impact of chromosome evolution on mutation and recombination frequencies might result in directed evolution [135] and phenotypic changes. Additionally, lineage-specific loss of duplicated gene copies after gene duplication [136] or possibly evolutionary divergence of expression, as observed in European grayling and Atlantic salmon [137], may contribute to the formation of new species.

The activity of mobile elements with lineage-specific changes, such as those observed between Atlantic salmon and European grayling, is a primary driver of genome evolution [138] and could have also been engaged in numerous genome evolutionary changes. Furthermore, it has been claimed that chromosomal inversions, such as those often seen in the European grayling genome, have a significant impact on the processes of adaptation and speciation [139]. For instance, they enhance genome sequence divergence between saltwater and freshwater clades of the stickleback *Pungitius pungitius* [140], as well as non-migratory and migratory Atlantic cod ecotypes (*Gadus morhua*) [141,142]. Computer models supported these findings and demonstrated how chromosomal inversions could hasten speciation, particularly if adaptability includes several genes with tiny individual adaption effects [143]. A study on domestic mice (*Mus musculus domesticus*) revealed the possibility of fast differentiation driven by Robbertson fusions [144].

### 2.4. Local Genome Features of Different Salmonids

The literature found makes it clear that the genome of many salmonids has been studied in fish inhabiting specific local water bodies and rivers, therefore possessing their own uniqueness. Typically, one study will focus on a single population inhabiting a particular water body. However, this tends to reveal both the distinctive characteristics of individual salmonid populations and their relatedness. This section, therefore, focuses on examples from studies of some of the local groups that inhabit local watercourses and rivers.

Kanjuh et al. (2020) set out to map coho populations in poorly studied and unexplored rivers in the Danube basin in Croatia and to study their molecular diversity, gaining insight into phylogeographic haplotype patterns using different molecular markers [145]. Kalayci et al. (2018) suggested that all brown trout populations belonging to the Danube lineage should have nominate species status [146]. This is consistent with the estimate that diversification (i.e., genetic differentiation) between lineages should be high enough for absolute reproductive isolation between them [147,148,149]. Conservative morphology and low levels of diversity between trout taxa from different phylogeographic lineages on continuous morphological traits also confirm their close evolutionary relationships [150]. Initially, low genetic variability was observed within the Danube lineage [151], but thanks to numerous studies covering many populations, records now show higher values of genetic diversity [152,153,154]. Simonovic et al. (2017) found that *Da1* is the most widely distributed local haplotype of coho salmon [155]. However, an anthropogenic factor is also considered important for their spread beyond their natural range, primarily because of the long tradition of the stocking with factory fish of AT origin that exists in many countries [156]. Similarly, mainly due to the activities of fishermen, wild trout of the AT line has been introduced in suitable conditions in Croatia [157]. Thus, the stocking of AT line trout outside the territory of the Republic of Croatia has been recognised as the main reason for the loss of native genetic diversity of the genus *Salmo* spp. [150]. A disjunctive distribution of haplotype *Da22* allowed Simonovic et al. (2017) to hypothesize that the coho salmon populations still possessing this haplotype are remnants of a once widely distributed population, probably of Late Miocene and Early Pliocene Ponto-Messinian age [155]. The same authors believe that the recent/modern molecular diversity of coho salmon in the area is the result of their dynamic evolution. In the middle section of the Una River, the *Da22* haplotype was accompanied by the *Da2* and *At1* haplotypes, both of which were considered non-native and were probably introduced into the population by the stocking of factory-grown coho salmon [158].

The mixing of these home-run trout with local populations has caused introgressive hybridization and reduced the genetic integrity of local trout so that in some geographical areas, the original populations have been almost completely mixed or replaced by non-native trout [159,160]. This context of phenotypic plasticity and ecological adaptation has contributed to a confusing nomenclatural pattern of Italian native trout with many morphs described as species (or subspecies) whose validity is questionable [161]. Rossi et al. (2019) previously carried out work to identify and assess the genetic composition of residual local populations of Mediterranean trout from different sampling locations in the Italian Lazio region, for which they applied two molecular markers for the diagnostic origin and geographic origin of trout. The main similarity of Italian trout is that they are part of the *Salmo trutta* complex, defined on a molecular basis [156]. The results obtained by Rossi et al. confirm the harmful effects of the mass introduction of domestic trout into Italian rivers already observed in other geographical regions of Italy [159,160,161] or Latium [162], with some exceptions specific to some island territories [163,164]. The most frequent AT haplotype (*At18*) corresponds to the haplotype originally identified in samples from Norway, Denmark, and Spain [165,166]. These data show that the practice of stocking domestic trout has some effect, spreading among allochthonous genotypes [160]. Atlantic trout have almost completely displaced native Mediterranean trout at some sites or at least mixed with them [167].

Interesting results were also found in some regions of Russia. Gordeeva et al. (2018) analyzed sequence variability of the mtDNA control region (537–547 bp) in 25 populations of European Russia and Siberia [168]. Currently, the Siberian part of the *C. Albula–C. sardinella* range is relatively isolated from the European part; however, none of the studied genetic markers (allozymes and different mtDNA sequences) show interspecific differences between grouse and grouse populations. The introgression of northern Dolly Varden haplotypes (Bering phylogenetic group) in the Arctic char genome has been found in both North America [169,170] and Asia [171,172,173,174]. Furthermore, analysis of the nuclear sequences *of rag1* did not reveal any substitutions strictly specific to *C. albula* or *C. sardinella.* Moreover, populations in which some fish have morphological and genetic features of *C. vandesius*, and others have features of cisco occur in a wide area covering water bodies of the Pechora River basin and adjacent regions of Eastern Europe [175]. The authors also discovered the haplotype *BER10* in the lower Anabar River; this finding shifts the boundary of its distribution along the Siberian Arctic coast further westward. It should be noted that many other species native to northern ecosystems with a wide range were originally described in different parts of the range as separate taxonomic units [176,177]. However, a thorough examination of these animals’ morphological, ecological, and genetic characteristics, particularly those occupying overlapping subspecies/species ranges, frequently finds species with intermediary morphological and genetic characteristics, raising questions about their authenticity [178,179,180,181]. The presence of the *SIB25* haplotype in Lake Siysk from the same area indicates that the ranges of Eurasian and Beringian phylogenetic groups overlap there, as in the lower Lena and Olenek. The introgressive hybridization with Dolly Varden could have occurred in the Pacific basin during one of the glacial maxima; from there, the Arctic loach with the Bering group haplotype dispersed along the Arctic Ocean coast during the subsequent climate warming. One can also acknowledge the former westward expansion of the chrysalis itself and its hybridization with the Arctic char in the Arctic regions of Siberia, with its subsequent disappearance. Either way, the evidence suggests that the Beringian cutthroat participated in the colonization of the eastern Arctic coast of Siberia, and the presence of only one haplotype over a vast area from Anabar to the Indigirka delta indicates that their ancestors passed through a narrow neck [168,182].

Thus, we have reviewed the main questions and points that have been asked in salmonid genome research to date (Table 1).

**Table 1 genes-13-02221-t001:** Data on salmon genomes ever obtained. As we can assess, the great majority of studies focus on the mitochondrial genome.

View	Genome	Genome Data Acquisition Approach	Characteristics and Features of the Genome	Reference
*S. alpinus erythrinus*	Mitochondrial	Sanger sequencing and annotation by comparison with other sequences using Geneious R11 Genome structure determined using MEGA X	Size: 16,652 bp; two ribosomal RNA (rRNA) genes, 13 protein-coding genes, 22 tRNA genes; the overall base composition was 28.0% A, 26.4% T, 17.0% G, and 28.6% C, and the mitogenome GC content was 45.6%	[183]
*S. alpinus alpinus*	Mitochondrial	Primers were designed using the mitoPrimer_V1 program; sequences annotated by comparison with published charr mitogenome sequences using Geneious R11	Size: 16,655 and 16,657 bp; GC content: 45.6%; 24 SNPs. The highest was the variability of NADH dehydrogenase subunit genes (42.6%Table of all variable sites).	[184]
*Salvelinus taranetzi*	Mitochondrial	The sequenced fragments were de novo assembled into a complete mitochondrial genome and annotated by comparing with published genome sequences of charr using Geneious R11	Size: 16,654 b.p. The overall base composition was 28.0% A, 26.4% T, 28.6% C, and 17.0% G, with a slight A + T bias (54.5%). Two substitutions were found in the control region and 12S rRNA. Other single nucleotide substitutions were found in common protein-coding sequences The total sequence discrepancy (D xy) was 0.0011 ± 0.0002	[185]
*Salvelinus* sp.	Mitochondrial		Size 16,654 b.p. The organization of the genome was identical to that of typical salmon genomes, including 2 rRNA genes, 13 protein-coding genes, 22 tRNA genes, a light chain origin of replication (OL), and a control region (CR). The overall base composition was 28.0% A, 26.4% T, 17.0% G, and 28.6% C, and the GC content was 45.6%.	[186]
*Salvelinus boganidae* and *Salvelinus elgyticus*	Mitochondrial	Libraries were prepared using an Ion Plus Fragment Library Kit and unique adapters (Ion Xpress), and preliminary fragmentation of PCR products was performed on a Covaris M220 ultrasonicator. Libraries were sequenced on the Ion S5 platform (Thermo Fisher Scientific). Clean reads were assembled into contig with the Bowtie2 algorithm in Geneious R11 Mitogenome of *S. taranezi* was used as a reference for correct contig position and orientation	16,654 b.p. for *s. elgyticus* and 16,655 b.p. for *s. boganidae*. The gene arrangement, composition, and size are similar to *s. Taranezi* The genomic organization is identical to typical salmon mitogenomes, including two rRNA genes, 13 protein-coding genes, 22 tRNA genes, a light chain origin of replication (OL), and a control region (CR).	[187]
*Salvelinus levanidovi*	Mitochondrial	Using Geneious R11, the sequenced fragments were combined into a full mitogenome and described by reference to previous char mitogenomes.	The complete mitogenome of native *S. levanidovi* was 16,624 bp long. the overall base composition was 28.1% A, 26.4% T, 28.6% C, and 16.9% G, with a slight A + T bias (54.5%)	[188]
*T. thymallus*	Mitochondrial	The consensus sequence was controlled and annotated using MitoAnnotator.	Mitogenome has a total length of 16,660 bp and includes 13 protein-coding genes, 22 transfer RNA genes including 2 tRNA-Leu and 2 tRNA-Ser, two ribosomal RNA genes, and a control region following the standard vertebrate order. Intergenic spaces and overlapping sequences were found. Six coding genes have an incomplete codon stop: NADH2, COII, ATP6, NADH3, NADH4, and *cytb*. The base composition of the entire genome was 27.5% for A, 27.9% for T, 17.5% for G and 27.1% for C.	[189]
*Sl. Svetovidov*	Mitochondrial	Genomes were matched using the MAFFT algorithm in Geneious, and maximum likelihood (ML) analysis was performed based on Tamura-Nei (TrN93) plus a nucleotide substitution gamma model. The machine learning tree was built using the MEGA X software and validated by bootstrap analysis.	Genome length 16,655 bp. The overall base composition was 28.0% A, 26.4% T, 28.7% C, and 16.9% G with a slight A + T bias (54.5%) 18 single nucleotide differences and no difference in length between sequences MK695627, MK695628 and MK695629; only 13 substitutions were found in all sequences encoding the protein and five were found in the control region.	[190]
*O. masou*	Mitochondrial		Genome length 16,648 bp, with 13 protein-coding genes, two rRNA genes, 22 tRNA genes, a putative control region (CR), and one light chain origin of replication. The overall base composition is 28.6% A, 26.8% T, 28.1% C, and 16.4% G, respectively, with a slight AT bias (55.4%).	[191]
*S. fontinalis × S. malma* sp. hybrid	Mitochondrial		Genome length 16,623 bp. The overall base composition was 28.3% A, 26.5% T, 28.4% C and 16.8% G, with a slight A + T bias (54.8%) two non-coding regions, the L-chain origin of replication was located between tRNA-Asn and tRNA-Cys, and the control region was located within tRNA-Pro and tRNA-Phe. Eight tRNAs and the ND6 gene were encoded on the L-chain, the rest were encoded on the H-chain The complete mitogenome sequence contained 16s RNA and 12s RNA, which were located between tRNA-Phe and tRNA-Leu and were separated by the tRNA-Val gene.	[192]
*S. malma*	Mitochondrial		Genome length 16,652 bp. The overall base composition was 28.1% A, 26.4% T, 28.5% C, and 17.0% G. The 13 protein-coding genes code for 3808 amino acids in their entirety.	[193]
*S. trutta*	General	Sequencing with Illumina HiSeqX; Assembly of the genome using a software package; Gene annotation with Ensembl	Size: 2.37 Gbp-p; most of the assembly consists of 40 chromosomal pseudomolecules; 43,935 protein-coding and 4441 non-protein-coding genes	[194]

## 3. Phylogenetic Characteristics of Salmon

The number of teleost species far outnumbers any other group of fish or any other vertebrate, and this has been associated with a whole genome duplication prior to their emission during the Cretaceous [27,195,196,197,198]. Salmonids present a unique opportunity to explore a range of evolutionary and ecological concepts, including the mechanisms of speciation [199,200], the evolution of complex life cycles [197,201,202,203,204], patterns of chromosomal evolution [101], and genome duplication [40], as well as the role of hybridization in evolution [205]. The WGD mentioned above has played an important role in the long-term evolutionary success of salmonids by providing lineage-specific physiological adaptations such as anadromy, therefore potentially promoting evolutionary diversification and speciation [70]. According to the same authors, most of the salmon genome has returned to a state of diploid inheritance before subfamily divergence [70]. In this regard, salmonids occupy a unique phylogenetic position compared to fish species whose genomes have been or are in the process of being sequenced since they belong to the *Protacanthopterygii*, the most primitive group of teleosts [206]. Thus, salmonids provide a key phylogenetic link between the evolution of teleost fish and the evolution of non-bony fish, as well as other vertebrates. Salmonid evolutionary connections have been the focus of considerable systematic and phylogenetic investigations for several decades [207,208,209,210,211,212,213,214,215,216,217,218,219,220,221,222,223]. The derivation of a reliable phylogeny for this group is important for comparative analysis of salmon adaptation [224,225,226], comparative genomics [227,228], and for assessing conservation priorities [38,47,215,229]. Population genomics is now commonly applied to salmonids without genome sequencing by using conserved synteny with rainbow trout or Atlantic salmon, for example [38,47,48,229,230,231].

Salmonidae is a phylogenetically distinct family that arose, according to the latest data, from 50.8 to 64 million years ago [40,232,233,234,235], and this is confirmed by morphological comparison data of 11 genera belonging to three subfamilies: *Coregoninae* (ciscoes, whitefish, and inconnu), *Thymallinae* (grayling) and *Salmoninae* (huchen, lenok, trout, char, and salmon) [211,236,237,238]. The most numerous of these, *Salmoninae*, includes seven genera distributed throughout the world: *Brachymystax* (flax), *Hucho* (huchen and taimen), *Oncorhynchus* (Pacific trout and salmon), *Salmo* (Atlantic salmon and trout), *Parahucho*, *Salvethymus* and *Salvelinus* [43,140,217]. There is strong evidence that each is a monophyletic clade, a natural group that includes all the descendants of its last common ancestor [75,239,240]. *Coregoninae* is divided into *Coregonus*, *Prosopium,* and *Stenodus*. The *Thymallinae* has one genus, *Thymallus*. Three of these genera (*Parahucho*, *Salvethymus*, and *Stenodus*) are monotypic, and their precise location in their respective subfamilies is being debated. Originally, *Salmoninae* phylogenetic rebuilding has permitted *Parahucho perryi* to also be found in a variety of positions within the group, including sibling *Salvelinus* [218,241], Salmo [158,223], and the rest of *Salmoninae* [232,242,243]. Based on these data, it has been suggested that *Coregoninae* is a sister group to the rest of the *Salmonidae* [217,244]. Recent genetic investigations of the subfamily *Thymallinae* utilizing genome-wide mitochondrial genome sequencing confirmed the relatedness of the three subfamilies *Coregoninae*, *Thymallinae*, and *Salmoninae* but left confusion regarding the evolutionary connections between the long-recognized subfamilies [244,245]. Other monotypic genera, such as *Salvethymus* in the *Salmoninae* and *Stenodus* in the *Coregoninae*, have distinct morphologies and karyotypes that set them apart from other genera in their respective subfamilies, although certain molecular data show that they possess not justify genus classification [185,218,236,241,243,246,247,248]. Comparison of currently obtained mitogenomes with 27 related-group mitogenomes available from GenBank, including the genera *Salvelinus*, *Parahucho,* and *Salmo*, indicates a close relationship of *S. alpinus erythrinus* to related species, *S. taranetzi* (and closely related taxa). The study of chromosome patterns served to decipher the pathways of evolution [249,250] and genome duplication [251,252]. A theory of tetraplodization of salmon as a result of exposure to radiation exposure between 25 and 100 million years ago was proposed; this period is a distinctive feature of the family [48,253].

Unclear issues concerning *Salmonidae* interactions could be clarified by expanding the sample size of both taxa and characteristics. So, Crête-Lafrenière et al. (2012) produced one of the most significant contributions to the study of salmon phylogeny, choosing 63 species, more than twice the amount in use by Stearley and Smith [217], and completing a complete morphological investigation to present. They hypothesized that increasing the sample of taxa could separate long branches and that this would allow for a more precise establishment of phylogenetic relationships [232]. Also, in their opinion, an increase in the sample of taxa could be useful in estimating parameters for models of molecular evolution [254,255] and various types of phylogenetic tests, including rooting analysis [256] estimating the time of divergence [257]. According to their results, *Parahucho perryi* is a sister group (*Salvelinus*, *Oncorhynchus*), which was confirmed later [100], and *Stenodus* and *Salvethymus* cannot be highlighted as independent genera [246,248,258]. Crête-Lafrenière et al. (2012) employed a variety of genes to infer salmon phylogeny, in addition to expanding the number of species studied. Single gene phylogenies are intrinsically restricted in their capacity to reliably establish taxonomic relationships and are prone to random mistakes. As a result, they integrated gene sequences into a “supermatrix” to boost phylogenetic signal and node support. However, simultaneous analysis of linked gene sequences should be treated with caution, as biases can interfere with phylogenetic inferences due to strongly supportive clades erroneously grouped based on multiple substitution artifacts (e.g., heterogeneity of nucleotide composition [259,260], change in speed at different sites [261,262] and an increase in the sample of taxa [254,255,263,264].

### 3.1. Subfamily Salmoninae

*Salmoninae* became a sister subfamily of the *Coregonidae,* some 35.6 mya. *Hucho* and *Brachymystax* were found to be sister genera with a common evolutionary lineage that diverged earlier than other members of the subfamily [265]. Another example is the positioning of the monotypic genus *Parahucho* as a sister genus of Salmo with an average divergence time of 21.9 Ma. This placement also appeared in Crespi and Fulton (2004) [223] as well as Alexandrou et al. (2013) [266] but in contrast to several other studies that have either grouped *Parahucho* with Salvelinus or simply as a sister group to the *Oncorhynchus/Salvelinus* clade or *Oncorhynchus/Salmo* clade [230,232,233,267]. *Parahucho* and *Salvelinus* share an evolutionary lineage with a common ancestor with Oncorhynchus, with the genus *Salmo* placed between these latter three genera (*Oncorhynchus*, *Salvelinus, Parahucho*) and others (*Hucho, Brachymistax*). In Gong et al. (2017), within each genus *Salmoninae*, the phylogenetic relationships between species differed slightly from those previously published [191,265]. Within the genus *Salvelinus*, for example, the phylogenetic relationships of *Salvelinus fontinalis* and *S. leucomaensis* appeared similar in Ma et al. (2016) [268], where both belong to different evolutionary lineages, but various in Sahoo et al. (2016) [269] and Balakirev et al. (2016) [270], where they shared an evolutionary lineage. To further verify the newly-defined sequences and establish their taxonomic status in *Salmoninae*, phylogenetic trees, including all *Salmoninae* species available in databases such as GenBank, were constructed based on maximum likelihood analysis [271].

Identifying the evolutionary relationships of monotypic genera of the subfamily *Salmoninae* is rather complicated and identifying the relationship between *Salvelinus*, *Oncorhynchus* and *Salmo* has been a source of debate. Monotypic genera are simply one source of ambiguity in determining evolutionary connections inside the Salmonidae subfamily. As a result of genetic investigations, the lengthy classification of Oncorhynchus and Salmo as sister species has indeed been substituted by the group *Oncorhynchus* and *Salvelinus* [220,223,272]. The greatest issues with phylogenetic reconstructing inside the genera are centered on three more species-specific taxa, *Oncorhynchus*, *Salvelinus,* and *Coregonus*. In molecular investigations of *Oncorhynchus*, uncertain and contentious [223,273,274], and many issues emerge concerning the connection between Pacific trout, which is disguised by repeated hybridization [238]. Likewise, *Salvelinus* species frequently hybridize, resulting in discrepancies in phylogenetic analyses [217,275,276].

The long-standing definition of *Oncorhynchus* and *Salmo* as related species was divided into the *Oncorhynchus* and *Salvelinus* groups based on genetic studies [220,223,238,272,277]. Ninua et al. (2018) studied the morphology and mitochondrial phylogeny of five nominate trout species from the Western Caucasus. According to their data, trout from the Black Sea and Caspian Sea basins represent a monophyletic evolutionary lineage (matrilineal clade) distinct from trout from other parts of western Eurasia, including those from the Atlantic, Mediterranean, and Indian Ocean basins [278]. Earlier, Mari et al. (2014) analyzed grayling from the Kama and Ural basins and discovered previously documented haplotypes that form a sister clade with Scandinavian haplotypes when compared to accessible published studies [279,280,281,282]. Analysis of the data they presented implies that the Caspian and Scandinavian clades separated 0.33–0.92 million years ago, based on a replacement rate of 0.5% every million years. Despite the short number of examined populations and the small sample size, the Caspian haplotype polymorphism is equivalent to that of European groups [280,283]. Individuals from the Bugurla and Barangulovka sample sites had considerable genetic similarities, indicating extensive gene flow between these two proximal streams, giving rise to a single population [282]. The basal diversification within *Salmo* that separated *S. salar* from the bull-trout lineage occurred, according to different authors, from 9.6 to 15.4 Ma [232,234,266,267,284]. The separation between the widespread European *S. trutta* and the recently described species *S. ohridanus* and *S. obtusirostris* occurred between 3 and 9 Ma, in the Late Miocene or Pliocene [40,232,267,285].

#### *Salvelinus* 

All populations of Arctic char, including Taranian char, form a monophyletic group. Thus, until a recent study, several forms of char were described as separate species [222,286,287]. It was established that three species of loaches of the genus *Salvelinus* inhabit Lake Elgygytgyn, two of which are considered endemics of this water body. The first species is the long-finned loach *S. svetovidovi*, originally described in a separate genus *Salvethymus*; moreover, it was assumed that its age is comparable to that of the lake itself [233,258,287,288,289]. Another endemic species of this lake is the smallmouth char *S. elgyticus*. A third species of charr, *S. boganidae* charr, was thought to be identical to the form of the same name from Taimyr [289]. Although there is very little data on nuclear gene comparisons between Taimyr and Chukotka boganidae, mtDNA data suggest a polyphyletic origin [290,291,292].

Gonen et al. (2015) presented the first phylogenetic relationships of salmonids based on a RAD sequencing dataset, including five salmonid species and 3050 loci in the analysis, and found that the number of identified orthologous *Sbf I* RAD loci decreased as the evolutionary distance between species increased: several thousand loci were preserved in five salmonid species (~50 million year divergence), and several hundred were preserved in more distantly related bony species (~100–100 million year divergence). Most (>70%) of the loci identified between more distantly related species were of genetic origin [293]. Later, Lecaudey et al. (2018) presented the first phylogeny of salmonid fishes based on a large RAD sequencing dataset with an extensive sample of family taxa [233].

The use of molecular genetic approaches enabled us to transcend the restrictions placed by S’s parallel diversity in the morphological features of *S. alpinus* [294]. Current ideas about the phylogenetic and phylogeographic structure of Arctic char are based on studies by Brunner et al. (2001) of the diversity of the mtDNA regulatory region in different populations across its range, as well as in Dolly Varden (*S. malma*), a similarly-related species from the North Pacific. Five phylogenetic haplotype groups were revealed, one (Beringian) associated to the northern form of Dolly Varden and four (Arctic, Atlantic, Siberian, and Acadian) associated with Arctic char. Extensive research into charr mtDNA has substantially extended our understanding of their genetic diversity and evolutionary linkages in the Pacific basin [70,184,246,294,295,296,297,298,299,300,301,302,303,304,305,306,307,308], in Eastern Siberia [171,173], and in the North American Arctic [170]. A revision of the systematics of the genus has been proposed, involving consideration of monophyletic groups identified by mtDNA analysis as separate species, in particular, *S. taranetzi* [300], which includes Asian and North American loach with Arctic group haplotypes [288,291,300].

Brunner et al. (2007) recognized loach phylogenetic haplotype groups as clusters with strong initial support in their phylogenetic haplotype trees [291]. Loaches from North America’s Arctic regions, the Canadian Arctic Archipelago, Greenland, and the Asian Taranian loach were included in the Arctic group; loaches from Quebec and Maine were included in the Acadian group; loaches from Siberia, Finland, and Spitsbergen were included in the Siberian group; and loaches from Northern Europe, the Alps, the British Isles, Iceland, Greenland, Newfoundland, and Labrador were included in the Atlantic. The latter two groups were discovered as being the most closely linked, and they were joined with the Acadian group to form the Atlantic-Siberian-Acadian supergroup. Haplotypes from the fifth (Beringian) group were discovered in Arctic char after being discovered in the northern Varden river, which is thought to be the consequence of the previous hybridization between the two species [169,170,171,173,294,296]. As new data accumulated, the differences between the Atlantic and Siberian groups became increasingly blurred. The insertion of additional Siberian haplotypes caused the latter to disintegrate, and bootstrap analysis revealed no strong support for Siberian haplotypes or their big combinations [171]. Moore et al. (2015) achieved a similar finding for both the Siberian and Atlantic groups [170]. Also, Yamamoto et al.’s (2014) tree, which includes just Siberian haplotypes released by Brunner et al. (2001) [291,301], is divided into two clusters. One cluster combines with the Atlantic group cluster, although with poor bootstrapping support. The discreteness of the Atlantic and Siberian clusters seen by Brunner et al. (2001) might be attributable to both a paucity of Siberian material and sequencing errors [170,287].

Some of the forms that originated in this region are reproductively isolated [287,309,310]. Boganidae (*S. boganidae*) and Dryagina’s loach (*S. drjagini*) are described as different species [222]. Thanks to this study, boganids from lakes Elgygygytgyn and Lama are attributed to different phylogenetic groups of the Arctic char lineage and are not considered as a single species, *S. boganidae* [70,287,288]. These forms are now strictly separated, just like the Boganidae *S. boganidae* and the smallmouth *S. elgyticus* [286]. The results of the *RAG1* gene analysis also unambiguously point to the belonging of Lake Cherechenskii char to the Arctic char lineage. However, the two species of loaches did not form a supported clade on the phyloo graph; *S. boganidae* grouped with *S. taranetzi* from Chukotka, as previously suggested by some authors [287]. Oleinik et al. (2020) tested two alternative hypotheses about the origin of loaches in Lake Elgygygytgyn. The data presented are most consistent with the conclusion that two different in-time colonization events by ancestral lineages of Taranec shiners occurred in Lake Elgygytgyn during the postglacial periods. Despite the stability of the topology, contradictory signals or alternative phylogenetic histories were found within the Arctic phylogenetic group. Taking into account the previous genetic work of these species based on microsatellites [287], they suggested that the reticulations detected by the Neighbor-Net revealed several signals of hybridization events. The phylogenetic network showed that past hybridization between *S. boganidae* and *S. elgyticus* is possible, although no hybrids were observed between them [287]. *S. boganidae* also shows potential hybridization in the past with *S. taranetzi*. Among loaches of the genus *Salvelinus*, phylogenetic relationships and taxonomy are most problematic within the group that includes the Arctic loach *S. alpinus*, the trout *S. malma*, and closely related forms and species of loaches [284,288,311,312]. These two groups of char are often considered within two species complexes (*S. alpinus* complex and *S. malma* complex) [312] or one supercomplex *S. alpinus*-*S. malma* complex [243].

Gordeeva et al. (2018) found Holts with Atlantic subgroup haplotypes in Karelia, the Kola Peninsula, Novaya Zemlya, Polar Urals, and Taimyr. Their finds indicate a wide distribution of several Atlantic haplotypes. The Siberian subgroup borders the Arctic subgroup to the east of the Indigirka river basin; however, the location of this boundary and the extent to which their ranges overlap is uncertain. The Siberian subgroup is almost entirely represented by sedentary continental populations; anadromous and insular populations have not been noted so far, except for one anadromous population from the Novaya River, Khatanga Bay basin. Different sets of haplotypes are detected in different isolated regions of the Siberian group range. However, these locations are not used to categorize the haplotypes in the phylogenetic trees. Earlier, common haplotypes *SIB8* and *SIB10* were found in Transbaikalia and the Yana basin [171]. These findings support the shared origin and tight evolutionary relationships of the Siberian subgroup’s Arctic loaches from various parts of Siberia, as well as putative migrations between these locations during the Pleistocene glacial maximum, but not as extensive as in the Atlantic subgroup [168].

As can be seen, current ideas about the phylogeny of loaches are based on studying the variability of the mtDNA control region [171,291,301]. Oleinik et al. (2015) identified several phylogenetic groups uniting closely related species of loaches (Arctic, Atlantic, Siberian, Acadian, Beringian, Western Pacific, and Eastern Pacific). Their results show that specimens of *S. boganidae* and *S. elgyticus* belong to the aforementioned Arctic group, *S. taranetzi*. Originally described as a separate species, it later became synonymous with the Arctic char *S. alpinus taranetzi* [312]. There is still a lack of molecular data to support this view [313]. Most previous studies of *S. taranetzi*, along with other loaches, are limited to analyzing only short fragments of a few mitochondrial and nuclear genes [173]. Molecular data based on ten microsatellite loci and sequences of the mtDNA control region and cytochrome b gene [287], cytochrome b and cytochrome c oxidase I genes [232], and RAD sequencing support the assignment of *Salvethymus* to the genus *Salvelinus* [233]. In a phylogeny based on the mitogenomes of the loaches, *Sl. svetovidov* represents the latest branch that diverged after the main group of species (*S. fontinalis, S. leucomaenis, S. levanidovi, S. namaycush*). The authors in this study sequenced and described two complete mitochondrial genomes of *S. taranetzi* for further study and more precise phylogenetic analysis. *S. levanidovi* was phylogenetically placed alongside other loaches but showed substantial divergence from them. *S. fontinalis*, *S. levanidovi*, *S. leucomaensis*, and *S. namaycus* represent a basal group of species in terms of mitogenome diversity, each corresponding to a separate evolutionary lineage. As a result, Oleinik et al. (2020) revealed that *S. levanidovi* is highly correlated to the single origin of the genus *Salvelinus* [184].

To determine the taxonomic status of hybrid salmon, Zhang et al. (2019) reconstructed the phylogeny of this salmon stock with other natural salmon populations based on complete mitogenome sequences. The phylogenetic tree showed that hybrid salmon are related to *Salvelinus* and are more closely related to *S. fontinalis*, distinct from *S. malma* sp. The complete mitochondrial genome sequence of the *S. fontinalis* × *S. malma* hybrid provided an important data set for a better understanding of mitogenomic diversity and evolution of salmonid fishes, as well as new genetic markers for studying population genetics and species identification [192].

Analysis of the nucleotide sequences of the control region and the *cytb* mtDNA gene indicates that the “boganids” and smallmouth characters from Elgygygytgyn are young species whose origin is most likely related to the events of the last ice age [287]. Both species have Arctic haplogroup mtDNA haplotypes and are sister species that may have formed in the lake during the postglacial time. The results of the phylogenetic analysis of the nucleotide data of the mtDNA site *ATPase6*-*NADH4L* indicate an allopatric origin of the smallmouth and “richnid” loaches of Lake Elgygytgyn. This contradicts the results of the analysis of the *CR* and *cytb* genes, according to which their sympatric origin is more likely [287]. Despite the inconsistency in the available mtDNA data, data on the mtDNA control region [170,258,287,313] suggest that the level of nucleotide and haplotype diversity in the Taranec shin population in North America and Greenland is much lower than previously assumed [280] and slightly lower than in some Asian populations. Previously, based on an analysis of their own and published mtDNA data, the difference between the two studies is that the authors included smallmouth and "richnid" char from Elgygytgyn in the first subgroup [258], and in contrast, the *ATPase6-NADH4L* mtDNA site data divided the two species into different subgroups [289].

Introgressive hybridization across distinct lineages or taxa has been widely established for *Salvelinus* species as a result of previous secondary interactions [71,233,314,315], such as between *S. malma* and *S. alpinus*, which recently diverged from each other, about 1.5 million years ago, and have overlapping current distributions [315]. The presence of parallel edges in the network, on the other hand, does not guarantee hybridization, simply the probability of hybridization [316]. The results of the phylogenetic analysis by Oleinik et al. (2021) are consistent with earlier studies based on mtDNA fragments [232,246,287,301]. In comparison to earlier studies’ phylogenetic trees, their phylogenetic tree based on the mitogenomes of the char species grew more strong and more trustworthy. Longer DNA sequences have previously been proven to give an acceptable resolution of relatively high connections in fish [317]. A study by Oleinik et al. (2021) additionally showed that mitogenomes could make up a reliable phylogenetic tree and resolve relationships between closely related species of loaches [318]. On the basis of morphological similarity, the boganids of Chukotka and Taimyr were attributed to the same species [286,306]. However, the results of mtDNA analysis indicate a probable polyphyletic origin of allopatric populations of Boganidae [287,308]. Boganida charr’s morphological resemblance in this example might be a symptom of parallelism, which is defined as the autonomous development of shared traits among closely linked taxa during evolution [287]. Data on the *RAG1* gene confirm the monophyletic origin of Arctic charr and Dolly Varden complexes [242]. Results from Lecaudey et al. (2018) specifically show hybridization signals between *S. malma* from the Bering clade and *S. alpinus*. An earlier study showed *S. alpinus* individuals with introgressed haplotypes from the Bering clade of *S. malma* along the eastern coast of Siberia to be parapatric [171]. In a very recent study, mtDNA introgression from *S. malma malma* to *S. taranetzi* near the Sea of Okhotsk, due to which representatives of the first species did not remain in this location completely [318]. In addition, postglacial hybridization between different glacial lineages of *S. alpinus* has been demonstrated to survive in separate refugia [170].

The mitochondrial gene relationship between *S. confluentus* and the rest of the *S. alpinus* group has not been revealed by nuclear genome data, confirming lots of evidence referring to introgression of the Arctic char’s mitochondrial genome in this species that might conceal the sister taxon’s affection with *S. leucomaenis* [319]. With the emphasis on the subfamily *Salmoninae* and the extensive coverage of taxa of the genus *Salvelinus*, the topology recovered from >28,000 loci is well-defined and well-supported by all methods used, providing clear answers to several phylogenetic uncertainties revealed by conflicting results of previous studies.

### 3.2. The Subfamily Coregoninae

#### *Coregonus* 

In whitefish, species identification in phylogenetic studies is complicated by parallel evolution [237,320,321,322], phenotypic plasticity [323], trophic population variability [324,325], and hybridization [326,327]. The greatest difficulty in identifying stems from the variability of the two phenotypes that distinguish whitefish and grayling, each of which was formerly thought to be a monophyletic subgenus [238]. Thus, within the subfamily *Coregonidae* about 33.1 million years ago, the genus *Prosopium* showed an evolutionary lineage that had previously deviated from the *Stenodus*/*Coregonus* lineage. *Coregonus nasus* and *C. chadary* had a common ancestor in the same evolutionary lineage that previously diverged from the other species [328,329]. After them, *C. clupeaformes* and *C. autumnalis* also showed a common and divergent evolutionary lineage [269,330]. Phylogenetic analysis using a maximum likelihood tree among 42 complete mitogenomes of the *Salmonidae* family and one Danio rerio sequence showed 99% sequence similarity to the *S. trutta* genome [269]. Nevertheless, Baikal omul, *C. migratorius*, possesses morphological characteristics that suggest it is whitefish, although it is more strongly linked to whitefish based on molecular similarities [331,332]. *S. trutta fario* L. has been phylogenetically placed with other *S. trutta* species, with obvious divergence [330]. Hence, the whitefish could be classed as a paraphyletic clade based on morphological categorization.

Graylings are also a paraphyletic group, according to molecular evidence [333], with the smallest grayling, *C. sardinella*, becoming more strongly linked to whitefish than other graylings. Previous research has demonstrated that *C. huntsmani* has a unique evolutionary lineage [334,335], although it does not keep this status in the genus. Various research studies have found parallels between *Stenodus* and *Coregonus* [232,336,337]. Crête-Lafrenière et al. (2012) found slightly different and reversing results. First, *C. huntsmani* was recognized as a sister species to the other *Coregonus* species; second, *Stenodus* is probably a separate genus rather than a subgenus within *Coregonus*. Species such as *Coregonus artedi*, *Coregonus hoyi*, *Coregonus kiyi*, *Coregonus nigripinnis,* and *Coregonus zenethicus*, according to their data, are not related as previously thought [338,339,340,341], while *Coregonus pollan* and *Coregonus fallalis*, by contrast, tend to be more conspecific [232].

The only species of another distinct lineage that have not left the lake after separating from the aforementioned two are Baikal indigenous whitefish, lake omul (*Coregonus migratorius*), and bottom lake whitefish (*Coregonus baicalensis*). In contrast to Europe and America, where morphologically similar species pairs are the consequence of postglacial secondary contacts between glacial isolates, sympatric limnetic-benthic divergence was repeated here several times within the same body of water over a long geological period due to Pleistocene fluctuations. According to *cytb* mtDNA, phylogenetic analyses indicated the interspecific linkages described for Baikal whitefish [342]. Comparative genetic ranges across the three lines of responsibility: Baikal endemics, North American lake whitefish, and European whitefish, imply comparable ages [343].

Another interesting representative of whitefish is muksun. The revealed low level of genetic differentiation of muksun and whitefish, along with the available literature data, indicates that they belong to the same species, *C. lavaretus*. Borovikova and Budin’s (2020) analysis of the genetic variability of muksun from the Khatanga River points to its polyphyletic origin. The presence of common haplotypes, on the one hand, indicates their divergence from a common ancestor within the region under study; on the other hand, each form includes representatives of phylogenetic lines originating outside it. However, analysis of the haplotype network suggests the presence of several large mtDNA phylogenetic lineages. The molecular genetic analysis clarifies the phylogenetic relationships of high-tip and low-tip muksun of the Khatanga River, as well as with other *Coregonus* species. It also allowed us to clarify the origin of these forms in the basin of this river. It turned out that the degree of genetic differentiation between the forms is low and does not exceed the intraspecific level. It is important that the intrapopulation differentiation of muksun is comparable to the level of differences between it and whitefish: 0.2–0.3 and 0.2–0.5%, respectively. It should be noted that the differentiation level of 0.5% is also typical for different whitefish populations. Besides, it should be noted that the polyphyletic origin of low stamen and high stamen density forms of muksun does not allow us to refer them to different taxa and raise their status even to the subspecies level. Obviously, muksun is characterized by independent morphogenesis in different water systems, as in the case of European whitefish [344,345]. The phylogenetic kinship of whitefish and muksun is evidenced by the presence of the same common sequence variants of all three genetic polymorphism markers used in this study (*ND1*, *COI,* and *ITS1*) [346].

The species of roach (*C. peled*) is genetically similar to the aforementioned vendace. Peled differs from *C. albula* and *C. sardinella*, first of all, by the number of chromosomes. Differences in the number and chromosomal arrangement of the nucleus organizer regions between the vendace and the peled confirm that they are separate species [347]. *C. peled*’s emergence as a species may be an instance of recent chromosomal differentiation that enabled reproductive isolation from *C. albula* and *C. sardinella*. The most compelling evidence that *C. peled* is a distinct species is the qualitative difference between its nuclear genome and that of *C. albula*/*C. sardinella* [348]. Nevertheless, for the sea bass (genus *Sebastes*), a similar scenario exists: *S. mentella* Travin, 1951 and *S. marinus* (Linnaeus, 1758) (*S. norvegicus* (Ascanius, 1772)) might be separated from each other [349]. As previously stated, the specimen discovered in Lake Sobachye was classified as *C. sardinella* morphologically but carried the *C. peled* haplotype *N42* [350].

Individual phylogenetic lineages might well be readily identified using median connective networks that indicate the taxon’s evolutionary history, although cladograms (dendrograms) are frequently useless. These species, like *C. peled, C. albula*, and *C. sardinella*, were not distinguishable by barcoding; nonetheless, haplotype analysis of another mtDNA sequence, D-loop, revealed that they were separate species [351]. The *nd1* sequence distinguishes *C. peled* from *C. albula* and *C. sardinella*, but its nd1 haplotypes form a single unique branch, unlike the nd1 haplotypes of both cisco and lesser cisco. Furthermore, we cannot rule out the possibility that this species is a descendent of hybridization between *C. sardinella* and *C. peled*, which might have taken place in the past because hybridization is prevalent in whitefish even in natural reservoirs [352,353]. Therefore, Borovikova and Artamonova (2021) observe that in the instance of mtDNA sequences that do not generally recombine, the most significant criteria is their monophyletic or polyphyletic origin rather than the difference between haplotype sets [180].

### 3.3. The Subfamily Tymallinae

The subfamily *Tymallinae* originated about 29.5 Ma [265]. The *T. burejensis* / *T. tugarinae* clade was the first to diverge, next by *T. yaluensis*, another that included *T. thymallus*, and the last one that included all other species of the genus. Despite these topological variations, *T. arcticus* and were sister species, as recently proposed by Liu et al. (2016) and Balakirev, Romanov, and Ayala (2017) [354,355]. Ma et al. (2016) showed that mitochondrial genomes could be a powerful marker for determining phylogeny within *Thymallinae*. Their study confirmed that Yalu grayling should be synonymous with Amur grayling (*Thymallus grubii*) at the whole mitogenome level [268]. Other studies addressing both the nuclear mitogenome [61] and the salmonid nuclear genome [232,268] proposed that *Thymallinae* and *Coregoninae* are sister subfamilies, while others studying mitochondrial [245,267] and mitonuclear DNA [230] have proposed *Thymallinae* and *Salmoninae* as sister subfamilies. Horreo’s (2017) study with the mitogenome of 46 different salmonid species contributed much to the aspect of phylogeny construction by mitogenome, asserts *Coregoninae* and *Salmoninae* as sister groups in the family *Salmonidae*, and provides new insight into the phylogenetic relationships between genera and species in this family, including the molecular nodes [265]. The *Thymallus* sequences are divided into two monophyletic groups. The first (including the newly sequenced mitogen) includes four nominal species: *T. thymallus*, *T. brevirostris*, *T. Arcticus*, and *T. baicalolenensis*. The second includes three nominal species *T. tugarinae*, *T. grubii*, and *T. yaluensis* [189].

Despite a significant amount of effort acknowledging various elements of salmonid phylogeny, several of their phylogenetic connections and evolutionary history remain unknown. The degree of resolution of these concerns varies at different levels of biological organization, from the proper position of the salmonid tree root to the significance of introgression in species or subspecies classifications [218] and systematic constraints (which include completely inadequate taxon or gene sampling) [223,224,225]. In future studies, the focus of this study on the genus *Salvelinus* should also be on the genera *Salmo* and *Thymallus*. In the case of *Salmo*, there is still considerable uncertainty regarding the evolutionary history of several known taxa, such as *Salmo marmoratus* (marbled trout) and *S. obtusirostris* (soft-snout trout), as well as *S. carpio* (carpione) [356] and other larger phenotypes over the entire range of the *S. trutta* species complex, all of which may have been involved in important hybridization events. The genus *Thymallus* requires a comprehensive molecular study in both East and Central Asia. This is due to the comparatively high species diversity in East Asia and the somewhat exceptional link between contemporary taxonomy and phenotypic diversity in Central Asia. More thorough full-genome investigations of individual taxa like *Salvelinus*, *Salmo,* and *Coregonus* could give highly helpful information on both evolutionary radiation processes and the distinguishing features of individual taxa for all salmonids.

Thus, observing trends in salmonid genetics and genomics research, we can notice that molecular genetic and phylogenetic studies reveal an understanding of the history of salmonids as a family. This allows us to identify salmonid ancestors and establish intervals of genetic variation in *Salmonidae*. The complexity of the duplicated salmonid genome causes difficulties in its study but, on the other hand, increases the duplicability and genetic diversity of this family. The use of genomic techniques in commercial fisheries is mainly viable and cost-effective; nevertheless, the conversion of genomic data into management strategies has halted [17,357].

## 4. Conclusions

Thus, by observing trends in salmonid genetics and genomics research, we can conclude that molecular genetic and phylogenetic studies reveal an understanding of the evolutionary history of salmonids as a family. This allows us to identify salmonid ancestors and establish intervals of genetic variation in *Salmonidae*. The complex duplicated genome of salmonids poses a challenge to study, but on the other hand, it increases the plasticity and genetic diversity of this family. Applying genomic approaches to fisheries management is mainly feasible and cost-effective [17]. Salmonid species provide exciting possibilities to research speciation and adaptation mechanisms within an ecological and evolutionary context. In particular, it provides an opportunity to study the effect of hybridization and genome duplication on the evolution of species. Thus, Thompson et al. (2020) discovered that the complicated migratory phenotype is the product of a single gene area, which will aid in the conservation and recovery of Chinook salmon [358]. In formulating fisheries management rules and recommendations, genomic technologies and their capacities for recognizing species, establishing management units, and monitoring natural resources should be thoroughly explored [359,360,361].

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
