# Peer review of "Salmonidae Genome: Features, Evolutionary and Phylogenetic Characteristics"

_genes, 2022, doi:10.3390/genes13122221_

Round 1
Reviewer 1 Report
This review entitled " Salmonidae genome features, its evolutionary and phylogenetic characteristics" attempted to provide a comprehensive overview of the current state of researches on the genomics and phylogeny of salmonid species and advance our understanding of the evolutionary history of Salmonidae. Overall I have only a few comments that should be addressed before being considered for publication.
1, The writing is not concise, especially don’t repeat yourself. For example, Line 75-78 and line 126-129 are similar to or exactly the same as line 97-100 and line 132-136, respectively.
2, Most of genome information of salmonid listed in Table 1 were mitochondrial genome, and only one general genome was listed. The authors neglected existing studies that already provided high-quality genome assembly for Salmonidae. Randhawa et al. (2021) reported that about 12 whole genome sequences (WGS) of Salmoniformes were available on the NCBI repository as of December (NCBI, 2020). Therefore, it remains doubtful that this study provides new resources or information for Salmonidae genome feature, which needs to be clarified more.
3, I noted that the references cited in line 1090-1095 were to published in 2004, 2008 and 2010. With the development and improvement of the sequencing technologies, there should be many researches on the phylogenetic relationships and evolutionary history based on genomic data, so the second limitation was unreasonable.
4, In abstract, the authors wrote “This review should get potential researchers to pay attention to the future perspectives of salmonid genomics, including the role of the novel technologies (such as genome editing) in elucidating the genetic and evolutionary traits that underlie functional variation in traits of commercial and scientific importance”. However, the authors didn’t describe this content in detail in the text. I think this content is important and interesting.
5, Line 1127-1128, the authors pointed that “translating genomic data into management practices has stalled” and the cited reference 17 was published in 2017. However, Thompson (2020) found that a complex migratory phenotype results from a single gene region will facilitate conservation and restoration of Chinook salmon. There are also other relevant papers in the past five years. This must bring into question the thoroughness of the literature searches of this review.
6, Be sure to use proper citation style.
Author Response
1. Corrected.
2. Thank you for emphasising this topic. Randhawa's et al. (2021) observations are undoubtedly informative and useful. However, the table given in our review represents published research data on achievements in obtaining salmonid genomic sequences, rather than databases that are already accessible, as Randhawa et al. provide (2021).
3. I couldn't see any references to studies dating back to these years in these lines. Could you please specify where they are located?
4. That is correct. We have indicated that our review seeks to draw attention to these issues by presenting current findings. Thus, we believe that a review of the current state of study of the genomics and phylogenetics of salmonids should also draw attention to the improvement of current methods and the incorporation of new ones.
5. Thank you for this comment. Indeed, there was an oversight on our part: as the review started to be written at the beginning of 2020, such a detail may have been missed and not further updated. Thanks to your comment, we have deleted the thesis you highlighted.
6. Corrected.
Reviewer 2 Report
Summary: This article aims to provide an overview of the state of Salmonid genomic and genetics, with a focus on the disciplinary-level value of this important species. The review article impacts on ecology, physiology and evolution - as viewed through the lens of its importance as an aqua/agri culturally relevant organism. Generally, the work is well written, although at times departs from Standard English - however, the authors claim of the potential value of this work - specifically to promote future work in this system - is well founded. I offer suggestions for improvement to the authors below.
L84: Protopterus annectens should be italicised. Please ensure conformity throughout the article so that references to Latin names, genes (see L92), mRNAs are all presented in italics.
L93: "such as phylogenetics" reads awkwardly, do you mean the value of gar fish for their phylogenetic position?
L109: The sentence beginning "In particular, " is entirely unclear to me. Do the authors mean population-level genomic sequencing? Please clarify the meaning in context.
Introduction: from L49 - L143 there is not a single paragraph break. Please correct this syntax error. Additionally, much of this summary is irrelevant to the topic in the paper, please trim for brevity.
L158 - ohnologueus? Please check spelling and/or concept. Different spelling in L199. Please ensure correct spelling of all terms throughout.
L377: homologous?
L407: delete the "'s"
L624: Should Section 3. not be presented earlier? This section reads more like the introductory portion of the review.
L836: "Some of the forms formed" is redundant/awkward - please reword
L907: "cytb" should be italicized, check throughout.
There are many very long stretches of text/narrative in the article that are not broken up into paragraphs. In addition to being poor syntax, this makes reading the information very difficult for the audience.
Overall, my principal - and most important critique - relates to the focus of the review. Although much of the information provided appears sound, the breadth of the information presented is unwieldy. Rather than focusing specifically on the salmonid genome and topics presented in the abstract, this review goes well beyond, addressing massive topics such as whole genome evolution, paralogues, karyotypic behavior, etc. etc. This is too broad for a review. The authors must re-think the information they are presenting, and focus their topics to precisely what is presented in the abstract. At present, this review reads more like a textbook and, in my opinion, is too broad. This breadth threatens the readability and potential impact of the review article. I will note, alongside this important critique, that the information that is presented on the Salmonid genome and relevant topics to the abstract are appropriate - it just needs to be trimmed overall to be more focused of a review.
Author Response
L84: Corrected.
L93: Corrected. This misunderstanding was due to a translation mistake.
L109: As you can see, this thesis is followed by a reference to a review on the impact of different genomics technology approaches on the effectiveness of fisheries management. Thus, it refers to different approaches, including NGS.
Introduction: from L49 - L143: Corrected.
L158: Corrected.
L377: Corrected.
L407: Corrected.
L624: The introductory part of the third section is a preamble to the issue of salmonid phylogeny specifically, to which the third section actually deals, but not to the whole review, aiming to consider both phylogeny and features of the genome.
L836: Corrected.
L907: Corrected.
Summary responce: Thank you for such a touching comment on the basis of our review. Indeed, we have subjected our review to a certain complex branching in the course of consideration of the questions directly related to genomics, in order to review the adjacent problems, which, in our opinion, are not unimportant here. For example, consideration of genomic duplication is impossible without addressing the topic of whole-genome evolution; the problem of ohnologues/paralogues is also integral if we discuss a genome affected by a watershed evolutionary event. Thus, we accept this observation of yours as worthy of our paper, but in view of the difficulty of choosing theses that could be shortened, we are reluctant to do so. Furthermore, any review can be read selectively and any related information available can be useful in drawing attention to these issues as well.
Reviewer 3 Report
In general this comprehensive review is well written and thorough. It will make a good contribution to the field since it pulls together a lot of information into one place. The authors should be complemented on the amount of review work.
I have a few minor edits/suggestions:
The authors should exercise caution when using acronyms at the beginning of a sentence (example line # 199). I would not consider WGD as a "well known and common acronym" for those outside the field.
The legend for Table 1 is incomplete and needs further explanation and a more comprehensive description. The format of the talbe probably should also be reworked since the entries contain a lot of description that could go into multiple columns instead of a paragraph in a single column.
Figure 1 need further description. The legend is not descriptive and requires further information about the phylogeny.
Figure 2 needs further description as well, particularly with description of the outgroups used etc.
In general, legends to figures and tables should have full, stand alone descriptions of what is being presented.
Author Response
Thank you for your complimentary but honest assessment of our review.
Here is our response, point by point:
1. The first time the term is mentioned, an indication is given in brackets as to what abbreviation it will be referred to by in the future.
2. The commentary to the table has been supplemented slightly. Regarding the data in the table, I would be grateful if you could suggest which information would be better divided into separate columns. in our opinion, the table is too wide in its current state and might look worse in the manuscript if it were wider.
3. Corrected (supplemented).
4. Corrected (supplemented).
Round 2
Reviewer 1 Report
I'm sorry that the authors didn't answer most of my questions carefully. My principal critique in the first round review was the thoroughness of the literature searches of this review.
1、As I mentioned earlier, most of genome information of salmonid listed in Table 1 were mitochondrial genome, and only one general genome was listed. The authors neglected existing studies that already provided high-quality genome assembly for Salmonidae. However, the author thought that the table given in our review represents published research data on achievements in obtaining salmonid genomic sequences, rather than databases that are already accessible, as Randhawa et al. provide (2021). Please consult related literature carefully, such as De-Kayne R, Zoller S, Feulner PGD. A de novo chromosome-level genome assembly of Coregonus sp. "Balchen": One representative of the Swiss Alpine whitefish radiation. Mol Ecol Resour. 2020 Jul;20(4):1093-1109. doi: 10.1111/1755-0998.1318.
2、The number 256,277 and 278 references were published in 2004, 2008 and 2010, respectively. With the development and improvement of the sequencing technologies, there should be many researches on the phylogenetic relationships and evolutionary history based on genomic data, so the second limitation was unreasonable. Please rewrite this part.
3、In abstract, the authors wrote “This review should get potential researchers to pay attention to the future perspectives of salmonid genomics, including the role of the novel technologies (such as genome editing) in elucidating the genetic and evolutionary traits that underlie functional variation in traits of commercial and scientific importance”. However, the authors didn’t describe this content in detail in the main body of this article. Please describe this content in detail in the main body of this article, instead of a brief description in abstract.
4、In the first round review, I had pointed that there are some papers with relation to translating genomic data into management practices in the past five years, and listed one relevant paper. The authors just deleted the thesis I highlighted, and didn’t update relevant information. Please rewrite this part.
Author Response
- In accordance with your remark, relevant additions have been made, namely the De-Kayne et al. (2020) study we have already cited was discussed in greater depth and is now reflected in our review to a greater extent. Also in the introduction, the availability of the NCBI database with the salmoniformes genomes obtained is now given.
- Corrected. This thesis has been removed in its entirety.
- We chose to correct this part of the abstract by highlighting what we are really writing about in our review. It is not so much about the current technologies for genome research themselves, but about the current state of research on this, which in turn should attract the application of new technologies.
- The message that "management practices have stalled" has been dismissed and the original idea supplemented with new data, including the research you cite.
Reviewer 2 Report
My principal critique from first review was that the manuscript is too broadly cast to retain relevance for the stated topic. I agree with the authors that some discussion of various features, e.g., genome duplication as a key event impacting paralogue identity/structure, are relevant. Importantly - my point is that they do not need to be quite so long. For the example provided, one can easily make the same point with several hundred fewer words. I maintain this critique and strongly urge the authors to scale down additionally for brevity as it will equate to a much more readable document for the audience, and thereby improve the impact.
Author Response
The first section of the review has undergone some reductions, which the authors found appropriate during a careful re-reading. Thank you for the remark.